

# Identification of Zip8-correlated hub genes in pulmonary hypertension by informatic analysis

FanRong Zhao[1,2,3,*], Yujing Chen[1,2,3,*], Yuliang Xie[1,2,3], Shuang Kong[1,2,3], LiaoFan Song[1,2,3], Hanfei Li[1,2,3], Chao Guo[1,2,3], Yanyan Yin[1], Weifang Zhang[1,4] and Tiantian Zhu[1,2,3]

[1] College of Pharmacy, Xinxiang Medical University, Xinxiang, China

[2] Xinxiang Key Laboratory of Vascular Remodeling Intervention and Molecular Targeted Therapy Drug Development, Xinxiang, China

[3] Henan International Joint Laboratory of Cardiovascular Remodeling and Drug Intervention, Xinxiang, China

[4] Departments of Pharmacy, The Second Affiliated Hospital, Nanchang, China

[*] These authors contributed equally to this work.

## ABSTRACT

**Background**. Pulmonary hypertension (PH) is a syndrome characterized by marked remodeling of the pulmonary vasculature and increased pulmonary vascular resistance, ultimately leading to right heart failure and even death. The localization of Zrt/Irt-like Protein 8 (ZIP8, a metal ion transporter, encoded by SLC39A8) was abundantly in microvasculature endothelium and its pivotal role in the lung has been demonstrated. However, the role of Zip8 in PH remains unclear.

**Methods**. Bioinformatics analysis was employed to identify SLC39A8 expression patterns and differentially expressed genes (DEGs) between PH patients and normal controls (NC), based on four datasets (GSE24988, GSE113439, GSE117261, and GSE15197) from the Biotechnology Gene Expression Omnibus (NCBI GEO) database. Gene set enrichment analysis (GSEA) was performed to analyze signaling pathways enriched for DEGs. Hub genes were identified by cytoHubba analysis in Cytoscape. Reverse transcriptase-polymerase chain reaction was used to validate SLC39A8 and its correlated metabolic DEGs expression in PH (SU5416/Hypoxia) mice.

**Results**. SLC39A8 expression was downregulated in PH patients, and this expression pattern was validated in PH (SU5416/Hypoxia) mouse lung tissue. SLC39A8-correlated genes were mainly enriched in the metabolic pathways. Within these SLC39A8-correlated genes, 202 SLC39A8-correlated metabolic genes were screened out, and seven genes were identified as SLC39A8-correlated metabolic hub genes. The expression patterns of hub genes were analyzed between PH patients and controls and further validated in PH mice. Finally, four genes (Fasn, Nsdhl, Acat2, and Acly) were downregulated in PH mice. However, there were no significant differences in the expression of the other three hub genes between PH mice and controls. Of the four genes, Fasn and Acly are key enzymes in fatty acids synthesis, Nsdhl is involved in cholesterol synthesis, and Acat2 is implicated in cholesterol metabolic transformation. Taken together, these results provide novel insight into the role of Zip8 in PH.

Corresponding authors
Weifang Zhang, z_weifang@163.com
Tiantian Zhu, zhutt@xxmu.edu.cn

## INTRODUCTION

Pulmonary hypertension (PH) is a syndrome characterized by marked remodeling of the pulmonary vasculature and increased pulmonary vascular resistance and pressure, leading to right heart failure and even death (*Hassoun, 2021*). PH pathogenesis is multifactorial and presented as the aberrantly elevated pulmonary artery pressure (PAP), the persistent increase in pulmonary vascular resistance, and the vascular remodeling (*Crosswhite & Sun, 2014*). The pathological features of the disorder include remodeling of the distal pulmonary vasculature, infiltration of inflammatory cells, and extension of the pulmonary artery smooth muscle cells (PASMC) into typically nonmuscularized vessels (*Hassoun, 2021*). A recent review pointed out that PH has been mitigated by drugs that improve vascular relaxation and inhibit cell proliferation, but the long-term effects are not ideal (*Mathew, 2011*). The newer drugs are urgently needed to improve survival and exercise tolerance.

SLC39A8 encodes a zinc transporter ZIP8, a member of ZIPs, whose expression was found to be highest in the kidney, lung, and testis, compared with other organs (*Wang et al., 2007*); in different organs, its expression was abundant in endothelium (*Tran et al., 2022*). The important role of ZIP8 in the lung has been demonstrated in several studies. For example, loss of ZIP8 expression was associated with impaired renewal capacity of type 2 alveolar epithelial cells (AEC2s) and enhanced lung fibrosis (*Liang et al., 2022*), and increased ZIP8 expression in lung epithelial cells was associated with a protective role against TNF-induced cytotoxicity (*Besecker et al., 2008*), and increased ZIP8 expression in the lung was associated with re-organization of filamentous actin (*Geng et al., 2018*). These data demonstrated the important role of ZIP8 in the lung.

A review has summarized the role of ZIPs in many diseases (*Takagishi, Hara & Fukada, 2017*), however, their role in vascular diseases had been paid little attention. Recently, some studies suggested that the expression of ZIP12 (another member of ZIPs) was induced in the vasculature in human patients and rat models of PH *in vivo* (*Tran et al., 2021*; *Xiao et al., 2021*; *Zhao et al., 2015*; *Zhu et al., 2022*), which was at least partially responsible for hypoxia-induced PH in both human and rats (*Zhao et al., 2015*). Other studies showed that ZIP14 could mediate the influx of $Zn^{2+}$ in sheep pulmonary artery endothelial cells (*Thambiayya et al., 2012*). Despite growing interest in the role of ZIPs in vascular disease, the understanding of the functions of ZIPs in PH remains a gap in current knowledge. Based on the above research background, we hypothesize that ZIP8, hereafter referred to as SLC39A8, could play a vital role in the progression of PH.

To test this hypothesis, we analyzed SLC39A8 expression in PH patients as well mice, identified four metabolic hub genes correlated with SLC39A8 by using bioinformatic methods, and validated the expression of these genes in PH mice, to predict SLC39A8-related molecular mechanisms in PH pathogenesis.

## MATERIALS & METHODS

### Data collection and processing

We searched the National Center for Biotechnology Gene Expression Omnibus (NCBI GEO) database for datasets related to both "pulmonary hypertension" and "Homo sapiens", identifying seven datasets. All the downloaded files were processed using R version 4.2.1, and the data were normalized, calibrated, and log2-transformed. Of these datasets, only those containing the expression of SLC39A8 in both PH patients and normal control (NC) samples were selected, such as GSE24988 (62 PH and 22 NC), GSE113439 (15 PH and 11 NC), GSE117261 (58 PH and 25 NC) and GSE15197 (18 PH and 13 NC). Considering the small number of specimens in each dataset, the 4 datasets were merged (153 PH and 71 NC), hereafter referred to as merged dataset, and followed by batch normalization using "sva (v3.32.10)" and "limma (v3.24.4)" R package to eliminate the batch effect.

### Identifying differentially expressed genes

Differential expression analysis was performed between PH lung tissue and normal tissue using the Limma package in R language. Genes were considered to be differential expression genes (DEGs) with $P < 0.05$ and $|\log_2 FC| > 0$. Results were visualized using "volcano" and "heatmap" plots constructed using "ggplot2 (v3.3.6)".

### Mouse model of PH/animal experiment

C57BL/6J male mice at 8–10-weeks were purchased from SPF (Beijing) Biotechnology Co., Ltd. Animals were randomized into two groups, kept at 20–25 °C under a 12 h light-dark cycle, and allowed free access to food and water freely. To induce the mice PH model ($n = 10$), received a single weekly subcutaneous injection of SU5416 (Su, 20 mg/kg body weight, suspended in carboxymethylcellulose solution). Then mice were housed in a hypoxic environment (10% $O_2$, Hx) for 4 weeks. Carboxymethylcellulose solution consists of four major components including 0.5% (wt/vol) carboxymethylcellulose sodium, 0.9% (wt/vol) sodium chloride, 0.4% (vol/vol) polysorbate 80, and 0.9% (vol/vol) benzyl alcohol in deionized water. Control mice ($n = 10$) received a vehicle instead of SU5416 and were subjected to normoxic conditions. A mean pulmonary arterial pressure (mPAP) of $\geq 25$ mmHg was regarded as a success of PH modeling, and researchers who tested mPAP were blinded to animal groups.

All animals survived until the end of the experiment. This study did not require euthanasia. At the end of the treatment, all mice were anaesthetized with pentobarbital sodium (30 mg/kg, i.p.) before being sacrificed, then lung tissue samples were collected for the subsequent experiments.

All experimental protocols were approved by the Ethics Committee of Xinxiang Medical University (XYLL 20230062) and administrated strictly following the Guidelines of the Laboratory Animal Center of Henan Province, Xinxiang Medical University.

### Reverse transcriptase-polymerase chain reaction (RT-PCR)

Total RNA was extracted from lung tissues using Trizol reagent (Invitrogen, Waltham, MA, USA) according to the manufacturer's instructions. The concentration of RNA was

**Table 1** The primer sequences of the interested genes in this study.

| Gene name | | Primer sequences | Product size |
|---|---|---|---|
| mouse Slc39a8 | forward | 5′- GGACTCGCTATTGGGACTCT-3′ | 287 bp |
| | reverse | 5′- GGGTTGGCATAGCAAGTCAC-3′ | |
| mouse Acat2 | forward | 5′-ACTTTCTGGGTGTAATTTTCTCTG-3′ | 382 bp |
| | reverse | 5′-TGGAATACCTACCCCACCTCA-3′ | |
| mouse Nsdhl | forward | 5′-AACCAGCAGTGCCAGTGTTGTC-3′ | 335 bp |
| | reverse | 5′-GGTGCTCAGCGGCTAAGATGTG-3′ | |
| mouse Acly | forward | 5′-CCAAGATCCCTGCAAGGAAAG-3′ | 205 bp |
| | reverse | 5′-TCAGGATTTCCTTGTGTCCCC-3′ | |
| mouse Fasn | forward | 5′-CGGCTGCGTGGCTATGATTATGG-3′ | 324 bp |
| | reverse | 5′- GGTTGCTGTCGTCTGTAGTCTTGAG -3′ | |
| mouse Gapdh | forward | 5′-CTTTGGCATTGTGGAAGGGCTC-3′ | 126 bp |
| | reverse | 5′- GCAGGGATGATGTTCTGGGCAG-3′ | |
| mouse Fdps | forward | 5′-CAGAGGAGCCTCGAGCATTTA-3′ | 240 bp |
| | reverse | 5′-GGAAGGCTTGTACCACGGTC-3′ | |
| mouse Acss2 | forward | 5′-TCCTGTCACCAAGCATAGCC-3′ | 343 bp |
| | reverse | 5′-AAACCGTGTGTGGTTCCCAT-3′ | |

determined with Nanodrop 1000 (Thermo Scientific, Waltham, MA, USA), and RNA (1 µg) of each sample was reverse transcribed using QuantiTect Reverse Transcription Kit (Qiagen). PCR was performed in technical triplicate for each sample by using a thermal cycler (GeneAmp PCR system 2400; PerkinElmer, Fremont, CA). Primer sequences used for the target genes analyzed are listed in Table 1.

## Gene set Enrichment Analysis (GSEA) of DEGs and SLC39A8-correlated DEGs

Gene set enrichment analysis (GSEA) was performed on the DEGs using GSEA/MsigDB (https://www.gsea-msigdb.org/gsea/msigdb). C2.cp.all.v2022.1.Hs.symbols.gmt (All Canonical Pathways) (3050) was selected as the reference gene set. Enrichment of gene sets was ranked according to normalized enrichment score (NES), which represents the strength of the enrichment and it denotes normalized enrichment score. Gene sets with |NES|>1 (positive NES scores indicate that gene set was upregulated in PH groups, negative NES scores indicate that the gene set was downregulated in PH groups), $P < 0.05$ and FDR ($q$ value) < 0.25 were considered significantly enriched (Zhou et al., 2020).

## Protein–Protein Interaction (PPI) network analysis and the hub genes identified

PPI network was constructed by Cytoscape software. In this study, STRING (version 11, http://www.webgestalt.org/) was used to analyze the PPI of SLC39A8-correlated DEGs, and interaction with a combined score >0.4 (medium confidence score) was considered statistically significant (Wu et al., 2021). Then, the PPI network was constructed and visualized using the Cytoscape software (v3.9.1) (Shannon et al., 2003). Hub genes were

identified using the Cytoscape plugin CytoHubba, and three different algorithms such as MCC, Degree, and Closeness were selected (*Feng et al., 2018*).

## Statistical analysis

Statistical analyses were performed using SPSS 18.0, GraphPad Prism 9.0, and R 4.2.1. Wilcoxon rank sum test and Welch $t'$ test was used to compare the difference of SLC39A8 and 7 key hub genes expression between PH and NC. Pearson correlation analysis was used to examine the relationship between the expression of SLC39A8 and other genes expressed in lung tissue of PH and NC. A Student's $t$-test was used to compare the difference in SLC39A8, Acat2, Acly, and Fasn expression in the lung between PH mice and control mice. Data were expressed as mean $\pm$ S.E.M.

## RESULTS

### Identification of differential gene SLC39A8 in PH and NC

As shown in Fig. 1A, a total of 5228 DEGs (($\log_2$FC > 0, $P$ < 0.05) were identified from the merged datasets (GSE24988, GSE113439, GSE117261, and GSE15197), of which 3,031 were downregulated and 2,197 were upregulated. The heatmap of DEGs in the pooled dataset showed hierarchical clustering of altered transcription in two groups (Fig. 1B), which may facilitate identification of the unknown transcripts' function or the unknown function of known transcripts.

Based on the GSEA analysis of all DEGs, the top 6 Kyoto Encyclopedia of Genes and Genomes (KEGG) pathways (Fig. 1C) included "Arrhythmogenic Right Ventricular Cardiomyopathy Arvc" (*Talati & Hemnes, 2015*), "Ecm Receptor Interaction" (*Thenappan et al., 2018b*; *Yuan et al., 2020*), "Wnt Signaling Pathway" (*de Jesus Perez et al., 2014*; *Konigshoff & Eickelberg, 2010*), "Hypertrophic Cardiomyopathy Hcm" (*Mitra et al., 2020*; *Musumeci et al., 2017*), "Focal Adhesion" (*Lin et al., 2017*; *Ravi et al., 2013*; *Zhao et al., 2014a*), "Dilated Cardiomyopathy" (*Dzięgwikecka et al., 2020*; *Liang et al., 2021*), all of which could be associated with PH. On the other hand, 4 of the top 5 enriched Reactome pathways (Fig. 1D) encompassed "Degradation of the Extracellular Matrix" (*Mumby et al., 2021*; *Thenappan, Chan & Weir, 2018a*), "Cilium Assembly", "Extracellular Matrix Organization" (*Thenappan, Chan & Weir, 2018a*), and "Signaling By Tgfb Family Members" (*Woo, Ornitz & Singh, 2019*), all of which could be involved in the pathogenesis of PH.

As shown in Fig. 1B, SLC39A8 was among the top 100 DEGs. To further verify its expression pattern, we next analyzed the expression of SLC39A8 in lung tissue between PH and NC. As shown in Fig. 1E, the expression of SLC39A8 decreased in the lung tissue of PH patients (Fig. 1E). To further validate this variation of SLC39A8 in PH, we examined the expression of SLC39A8 in a mice model of PH induced by SU5416/Hypoxia by quantitative RT-PCR. Likewise, SLC39A8 expression in the lung tissue of PH mice was noticeably reduced (Fig. 1F). Taken together, these results suggest that the datasets we screened are validated and SLC39A8 expression was downregulated in PH.

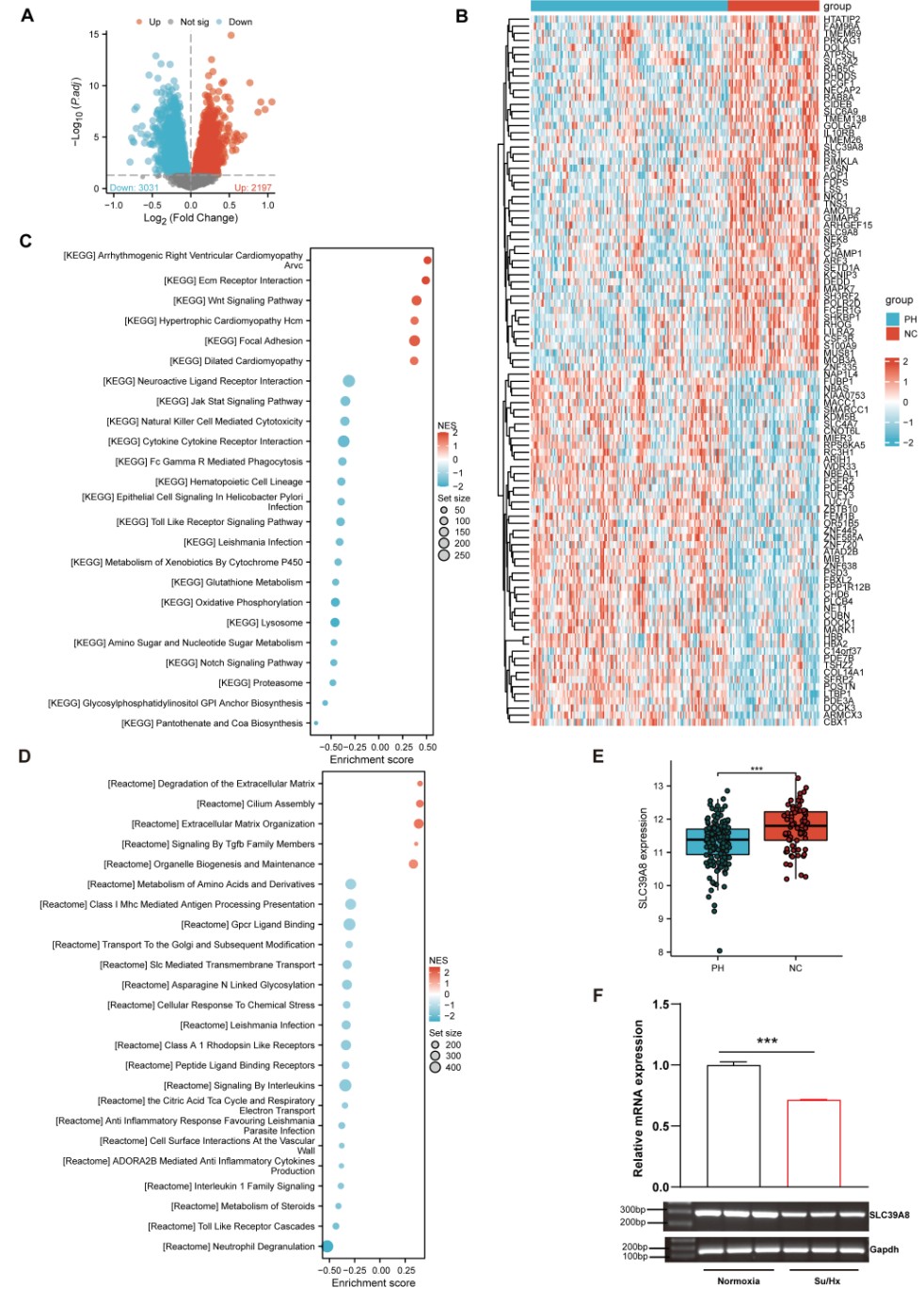

**Figure 1** **SLC39A8 expression was downregulated in PH patients and mouse PH models.** (A) The volcano plots of DEGs in the merged datasets (153 PH and 71 NC, merged from these four datasets), in which 3,031 DEGs were downregulated (log2FC > 0, FDR < 0.05) and 2,197 DEGs upregulated (log2FC > 0, FDR < 0.05). (B) Heatmap of DEGs in the merged datasets (C) GSEA analysis shows enriched KEGG pathways. (D) GSEA analysis shows enriched Reactome pathways. (E) Expression levels of SLC39A8 in PH and NC groups in the merged datasets. (F) Expression levels of Slc39a8 in the lungs of normoxia and Su/Hx treated PH mice ($n = 10$). Data were presented as mean ± SEM. ***$P < 0.001$.

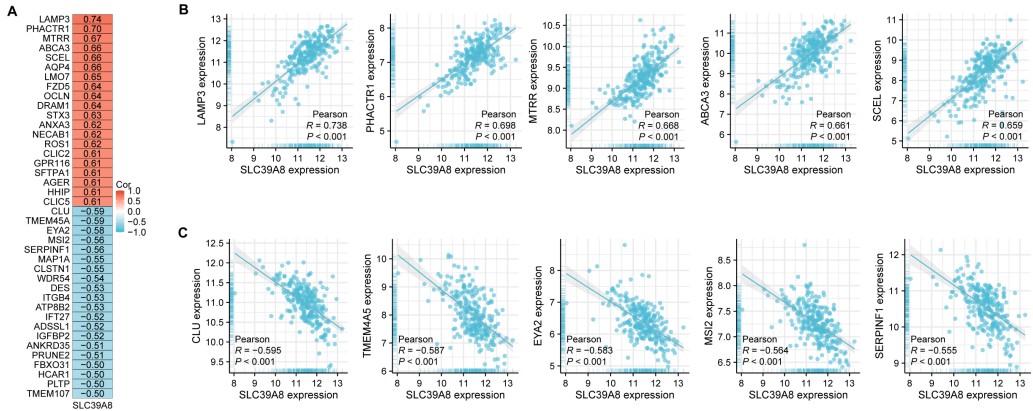

**Figure 2** **Correlation analysis of all DEGs and SLC39A8.** (A) Heatmap of the top 20 genes positively or negatively correlated with SLC39A8. Red represents positive correlation and blue represents negative correlation. (B) Top five genes positively correlated with SLC39A8 were displayed. (C) Top five genes negatively correlated with SLC39A8 were displayed.

## Identifying SLC39A8-correlated genes

To further explore potential molecular mechanisms of SLC39A8 in PH, we used Pearson correlation coefficients to perform correlation analysis on all genes expressed in PH and NC, thereby identifying genes with expression patterns correlated with SLC39A8 expression. Then 6,083 SLC39A8-correlated genes were identified when considered adjusted [adj.] *P* < 0.05.

According to the results of correlation analysis, 2,851 genes are negatively correlated with SLC39A8, and 3,232 genes are positively correlated with SLC39A8. As shown in Fig. 2A, top 20 positive-correlated genes and the top 20 negatively-correlated genes were displayed in heat maps of Pearson correlations. Then, the top five positively SLC39A8-correlated genes were presented in Fig. 2B, such as lysosomal associated membrane protein 3 (LAMP3), phosphatase and actin regulator-1 (PHACTR1), methionine synthase reductase (MTRR), methionine synthase reductase ATP-binding cassette transporter A3 (ABCA3) and sciellin (SCEL). Top five negatively associated genes were also presented in Fig. 2C, such as clusterin (CLU), Transmembrane protein family (TMEM45A), eyes absent homolog 2 (EYA2), Musashi2 (MSI2) and Serpin Peptidase Inhibitor, Clade F 1 (SERPINF1). Among them, CLU (*Liu et al., 2015*; *Nicolescu, 2015*) and ABCA3 (*Kunig et al., 2007*; *Ota, Kimura & Kure, 2016*) have been shown to be associated with PH, which indirectly suggests SLC39A8 may play a pivotal role in the progression of PH. Taken together, these results suggest that SLC39A8-related genes are associated with PH, which indicates that SLC39A8 may play a role in PH development.

## GSEA analysis of SLC39A8-correlated genes

After identifing SLC39A8-correlated genes, we further performed gene set enrichment analysis (GSEA) using gene set collections from the MsigDB (https://www.gsea-msigdb.org/gsea/msigdb/collections.jsp). These enriched pathways consisted of Wikipathways, Reactome, and KEGG. As shown in Fig. 3A, The SLC39A8-correlated genes were

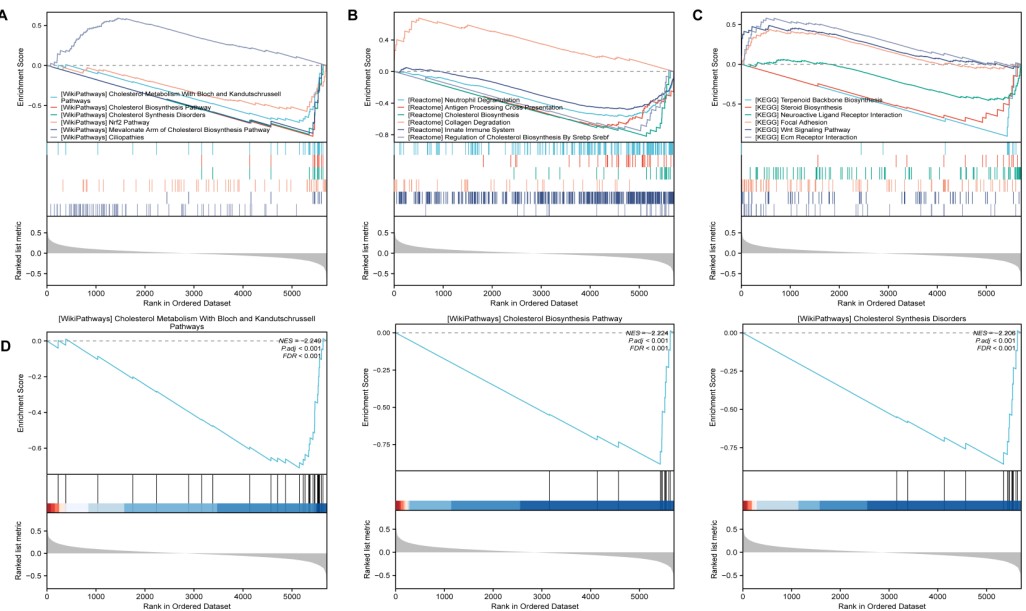

**Figure 3** **The GSEA analysis of SLC39A8-correlated DEGs between PH and NC.** (A) GSEA classical plots generated based on NES score in canonical Wikipathways. (B) The top three WikiPathways are listed respectively. (C) GSEA classical plots generated based on NES score in canonical Reactome pathways. (D) GSEA classical plots generated based on NES score in canonical KEGG pathways. $P$.adj < 0.05 and false discovery rate (FDR, qvalue) <0.25 were used to indicate significant enrichment score.

enriched in "Cholesterol Metabolism With Bloch and Kandutschrussell pathways", "Cholesterol Biosynthesis Pathway", "Cholesterol Synthesis Disorders", "Nrf2 pathway" and "Mevalonate Arm of Cholesterol Biosynthesis Pathway" in WikiPathways based on normalized enrichment score (NES). Among the top five WikiPathways enriched in the SLC39A8-correlated DEGs, four were clustered into a group "cholesterol metabolism", which was the most significant and has been demonstrated associated with PH (*Heresi et al., 2010*; *Jonas & Kopeć, 2019*; *Zhang et al., 2017*). Then, the top three pathways were displayed in Fig. 3B, respectively. According to the results of GSEA enrichment analysis of Reactome pathways, the SLC39A8-correlated genes were enriched in, "Antigen Processing Cross Presentation", "Cholesterol Biosynthesis", "innate Immune System" and "Regulation Of Cholesterol Biosynthesis By Srebp Srebf". Among the top five Reactome pathways, two were clustered into a "cholesterol metabolism" group. Furthermore, GSEA enrichment analysis of KEGG pathways revealed that the pathways enriched by SLC39A8-correlated genes included "Terpenoid Backbone Biosynthesis", "Steroid Biosynthesis", "Neuroactive Ligand Receptor Interaction", "Focal Adhesion" and "Wnt Signaling Pathway". Of these pathways, "Neutrophil Degranulation" (*Taylor et al., 2018*), "innate Immune System" (*Taylor et al., 2018*), Wnt signaling pathway (*de Jesus Perez et al., 2014*; *Konigshoff & Eickelberg, 2010*), and "Steroid Biosynthesis" (*Hester, Ventetuolo & Lahm, 2019*) have already been linked to PH. Collectively, the role of SLC39A8 in the progression of PH may be attributed to cholesterol and/or steroid metabolism, based on our results of GSEA analysis.
## Identifying key SLC39A8-correlated metabolic DEGs between PH and NC

Accumulating evidence has indicated that pulmonary arterial hypertension (PAH) is associated with metabolic dysfunction (*He et al., 2022*; *He et al., 2020*; *Wang et al., 2022*; *Xu, Janocha & Erzurum, 2021*). And a previous study has identified 1,660 human genes assigned to 86 metabolic pathways from the KEGG database (*Gong et al., 2021*). Based on these findings, we subsequently identified SLC39A8-correlated metabolic DEGs between the PH and NC, and constructed the PPI network of DEGs. As shown in Fig. 4A, a venn diagram revealed 202 common genes in three groups, including 1,660 transcripts of metabolic genes, 6,083 SLC39A8-correlated genes, and 5,228 DEGs between PH and NC. The 202 SLC39A8-correlated metabolic DEGs were analyzed by the STRING database to explore PPI, in which interactions with a combined score greater than 0.4 were obtained to construct PPI networks (Fig. 4B). The node color in PPI network changed gradually from yellow to red in increasing order according to the degree ranking of cytohubba (such as 1–2, 3–4, 5–7, 8–10, 11–19, 20–24, and 25–30). Hub genes were identified by the CytoHubba plugin in Cytoscape (https://string-db.org/). The top 15 hub genes were identified using three different algorithms closeness, degree, and MCC as shown in Figs. 4C–4E. A Venn diagram (Fig. 4F) revealed 7 common genes in the three groups, such as cholesterol acyl-transferase 2 (ACAT2), NAD(P)-dependent steroid dehydrogenase-like (NSDHL), farnesyl diphosphate synthase (FDPS), fatty acid synthase (FASN), ATP-citrate lyase (ACLY), farnesyl-diphosphate farnesyltransferase 1 (FDFT1) and Acetyl-CoA synthetase 2 (ACSS2), indicating that they represented the key SLC39A8-correlated metabolic DEGs between PH and NC. Together, these results further support the role of SLC39A8 in PH may be associated with cholesterol metabolism.

## Validation of key SLC39A8-correlated metabolic DEGs in a mice model using RT-PCR analysis

To further elucidate the relationship between SLC39A8 and the seven key genes. We presented the expression of these genes in PH patients and controls in the merged dataset and performed correlation analysis between these genes. As shown in Fig. 5A, PH patients lowly expressed six genes: ACAT2, NSDHL, FDPS, FASN, ACLY, and ACSS2. However, the expression of FDFT1 did not differ between PH and NC, this was indeed the case that the actual differences were small and not significant though there was a significant difference when differential analysis was performed, which attributed to different statistical methods used. Meanwhile, the correlation between SLC39A8 and these genes were shown in Fig. 5B. Interestingly, the expression of these genes was positively correlated with SLC39A8. Finally, we identified 6 genes that are most likely correlated with SLC39A8, such as ACAT2, NSDHL, FDPS, FASN, ACLY, and ACSS2.

To further verify the results of bioinformatics analysis, we examined the expression of these genes in PH mice and control mice. RT-PCR was used to validate the 6 key genes (NSDHL, ACAT2, ACLY, and FASN) expression in PH mice and control mice. As shown in Fig. 5C, the expression of Acat2, Nsdhl, Fasn, and Acly were significantly low in PH mice, which was consistent with the results of bioinformatics analysis. However, the expression of
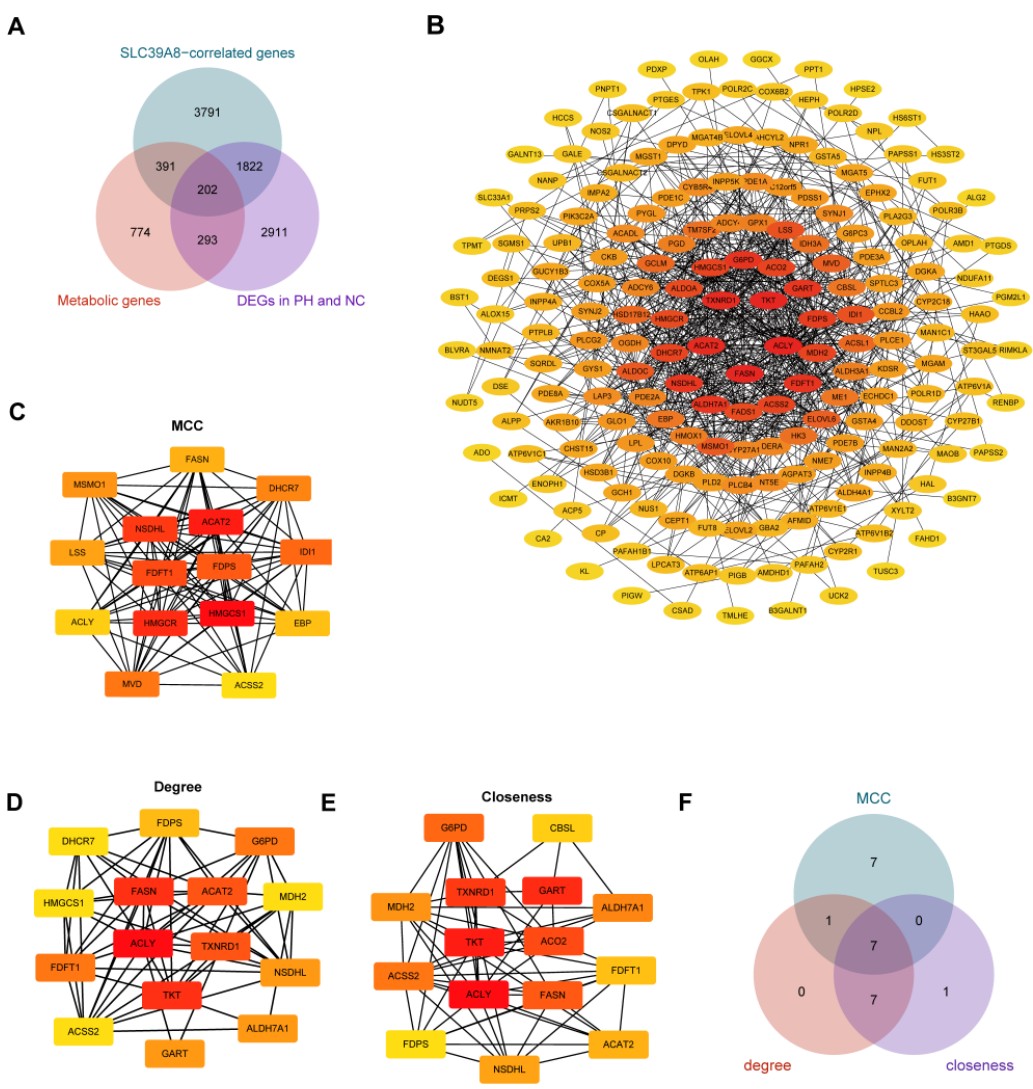

**Figure 4 Identification of hub genes SLC39A8-correlated metabolic DEGS between PH and NC.** (A) Venn diagram of common genes in three groups (1,600 transcripts of metabolic genes, 6083 SLC39A8-correlated DEGs, and 5,228 DEGs). (B) PPI network was constructed by the STRING database and visualized by cytoscape software (v3.9.1), and each blue filled node represents a SLC39A8-related gene; (C–E) The top 15 Hub genes were identified *via* cytoscape software (cytohubba) using MCC (C), degree (D), and closeness (E). (F) Venn diagram of common genes in these three hub gene sets.

Fdps and Acss2 did not differ between the two groups (Fig. S1). These data provided strong evidence that SLC39A8 expression in PH was associated with cholesterol metabolism.

## DISCUSSION

Pulmonary hypertension (PH) is a fatal rare disease that is characterized by pulmonary vascular remodeling, involving pulmonary artery endothelial cells, smooth muscle cells, and fibroblasts (*Humbert et al., 2019*; *Rabinovitch, 2012*). ZIP8 has recently been identified as a membrane transporter of essential and toxic divalent metals. The important role

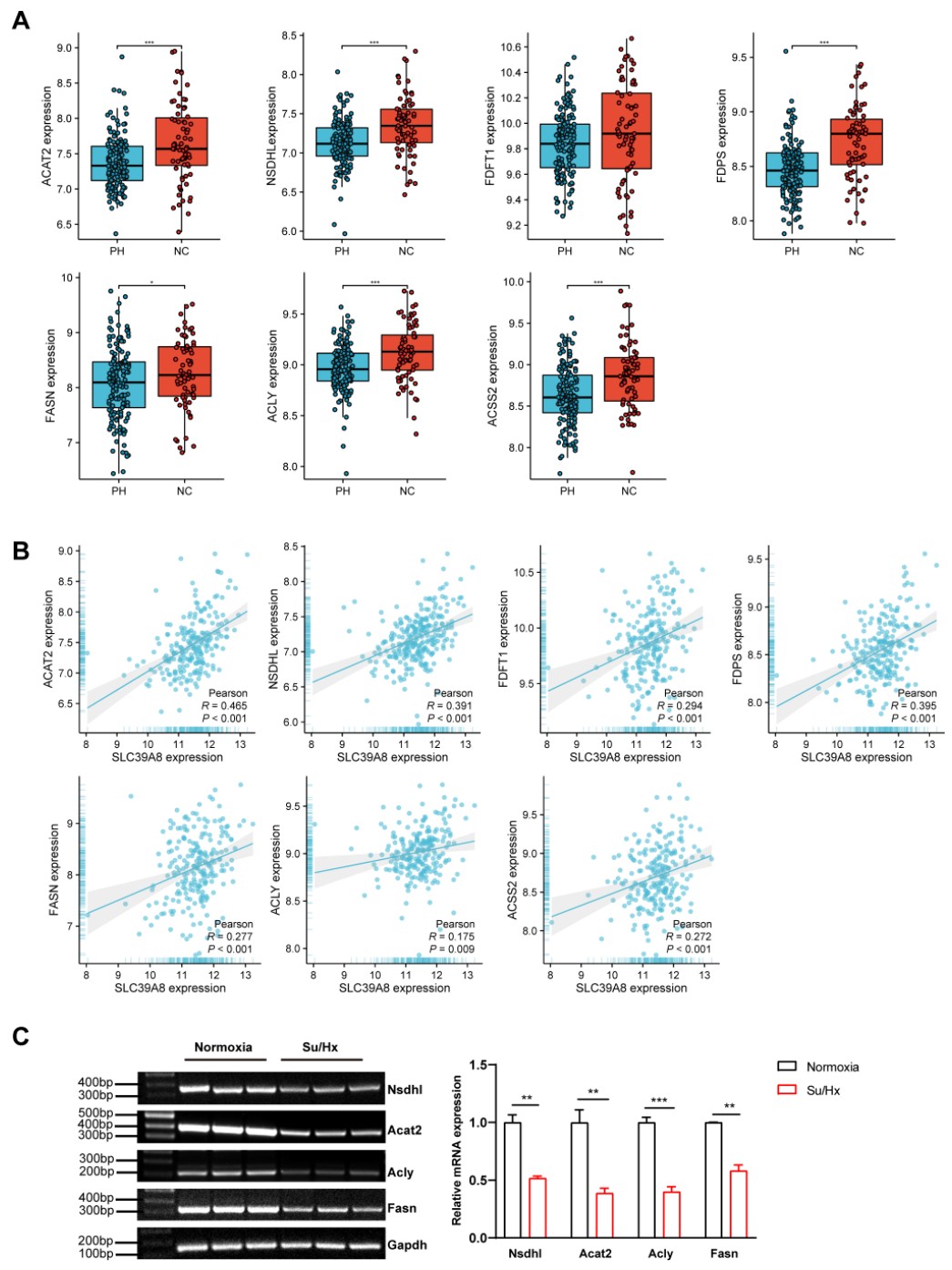

**Figure 5 Verification of hub genes expression at the mRNA level.** (A) The expression of 7 key SLC39A8-correlated metabolic DEGs in the merged dataset. (B) The correlations between 7 key SLC39A8-correlated metabolic DEGs and SLC39A8 were presented independently. (C) RT-PCR analysis of the expression of Acat2, Nsdhl, Acly and Fasn in lungs of normoxia and Su/Hx treated PH mice. $n = 10$ for each group. Data are shown as mean ± SEM; *$P < 0.05$, **$P < 0.01$, ***$P < 0.001$; ns, no significance.

of ZIP8 in the lung has been demonstrated, however, the role of ZIP8 in PH and the mechanism involved are unknown. In recent years, the rapid evolution of high-throughput sequencing technologies has provided a new perspective for PH research. The contribution of transcriptome technology in revealing the role of gene expression has long been appreciated.

In this study, we performed DEGs analysis between PH patients and controls, and found SLC39A8 expression was significantly reduced in PH patients. Furthermore, the low expression of SLC39A8 was confirmed in PH mice by using RT-PCR. These findings suggested that SLC39A8 may play a pivotal role in the progression of PH.

The results of GSEA analysis of all DEGs between PH and NC showed that the top five KEGG pathways were all associated with PH, and 3 of the top 5 Reactome pathways were also involved in the pathogenesis of PH. These results demonstrated the validity of the selected datasets. To further elucidate the potential role of SLC39A8 in PH, we identified SLC39A8-correlated genes in DEGs and performed a GSEA analysis of the correlated DEGs. The results of enriched Wikipathways, Reactome pathways, and KEGG pathways suggested that SLC39A8 was intimately linked to cholesterol metabolism.

Studies have shown that metabolic dysfunction is associated with PAH (*He et al., 2022*; *He et al., 2020*; *Wang et al., 2022*; *Xu, Janocha & Erzurum, 2021*). A recent review highlighted the role of obesity and lipid metabolism in the development of high-altitude pulmonary hypertension (HAPH), which suggests that triglycerides (TGs) and low-density lipoprotein (VLDL) could be predictors of HAPH in early stages, and high BMI is an important contributor to the development of HAPH (*Siques et al., 2020*). Furthermore, the role of imbalanced fatty acid metabolism in pulmonary arterial hypertension (PAH) also has been discussed (*Xu, Janocha & Erzurum, 2021*). It was interesting that body mass index (BMI) (*Speliotes et al., 2010*), obesity (*Berndt et al., 2013*; *Speliotes et al., 2010*) high-density lipoprotein (HDL) cholesterol levels (*Teslovich et al., 2010*; *Waterworth et al., 2010*; *Willer et al., 2013*) were correlated with rs13107325 SNP (results in Ala-Thr amino acid change at position 391 of the protein) of the Solute Carrier Family 39 Member 8 (SLC39A8) gene in several genome-wide association studies (GWAS).

Considered together, we speculated that SLC39A8 may play a role in PH by regulating cholesterol and/or lipid metabolism, and subsequently identified SLC39A8-related metabolic DEGs by using a Venn diagram. Next, seven hub SLC39A8-related metabolic DEGs were identified, after analyzing the expression of these genes and the correlation of these genes and SLC39A8, six hub genes were selected for further study. Finally, of the six hub genes, only four hub genes such as Acat2, Nsdhl, Fasn, and Acly were downregulated in PH mice, while the other two genes such as Fdps and Acss2 were equivalent between the two groups.

Of the four genes, NSDHL gene encodes a sterol dehydrogenase or decarboxylase enzyme involved in cholesterol biosynthesis (*Caldas & Herman, 2003*), ACLY is a key fatty acids synthesis enzyme, FASN is a key enzyme for the de novo synthesis of fatty acids, and ACAT2 is an ER membrane-spanning enzyme converting cholesterol and fatty acid to cholesteryl esters (CEs) (*Chang, Chang & Cheng, 1997*).

It is known that fatty acid metabolism involves fatty acid synthesis, fatty acid oxidation, and cholesterol metabolism (*Yang et al., 2019*). The importance of lipid mechanism in PH (*Siques et al., 2020*; *Xu, Janocha & Erzurum, 2021*) has been demonstrated, and imbalanced fatty acid metabolism is reported in the heart and lungs of PAH patients (*Hernandez-Saavedra et al., 2020*; *Zhao et al., 2014b*; *Zhuang et al., 2019*). A higher rate of de novo fatty acid synthesis was found in PAH-HPASMC, and increased expression of FASN was observed in the lungs of MCT-treated rats (*Singh et al., 2016*) and human PAH pulmonary arterial vascular smooth muscle cells (PAVSMC) (*Jiang et al., 2022*). In addition, another fatty acid synthesis enzyme ACLY was also upregulated in PAVSMC (*Jiang et al., 2022*). Furthermore, another study reveals that inhibition of FASN is beneficial for endothelial function in PH (*Singh et al., 2017*) and improves cardiac function associated with PH (*Singh et al., 2019*). These results demonstrated that the increased FASN is correlated with PH. However, in this study, our results indicated that the expression of FASN and ACLY in PH patients and PH mice were decreased. We speculate that the discrepancy might arise from the difference between the cell sample and tissue sample and need to be further studied.

Although cholesterol and fatty acids (FA) are essential lipids that play a wide range of physiological roles, excessive polar lipids, such as free cholesterol (FC) and free FA (FFA), are the major risk factors in the body. The stabilized ACAT2 converts cholesterol and FAs to CEs, thereby reducing the lipotoxicity of polar lipids. Previous studies have found HDL-cholesterol reduced (*Heresi et al., 2010*) in PAH patients, and loss of membrane cholesterol contributes to impaired pulmonary endothelial store-operated $Ca^{2+}$ entry (SOCE) in chronic hypoxia-induced PH (*Zhang et al., 2017*).

Those findings suggested the protective properties of HDL in PAH (*Jonas & Kopeć, 2019*). In this study, the expression of NSDHL and ACAT2 was decreased, which can decrease the production of cholesterol and increase the toxicity of cholesterol, finally participate in the procession of PH.

Collectively, the results of the present study revealed that SLC39A8 expression is low in pH patients and mice, we first identified its potential target genes associated with fatty acid metabolism through bioinformatic prediction, and validated the expression of these genes in PH patients and mice. Finally, we conclude that SLC39A8 may play a pivotal role in the progression of PH by regulating fatty acid and/or cholesterol metabolism.

However, some limitations should also be noted in this study. First, the sample size of the included datasets in this study is not big enough. Second, the specific regulatory mechanism between SLC39A8 and the 4 hub genes has not been explored. Therefore, a more detailed investigation of the protective role of SLC39A8 in PH and whether these four hub genes were involved will be required.

## CONCLUSIONS

Our data presented here was the first, to our knowledge, to show that the expression of SLC39A8 was low in the lung of PH patients by analyzing four publicly available microarray datasets retrieved from the GEO database. This result was validated in PH mice by using

RT-PCR. Furthermore, based on our current study, our research provided a bioinformatic analysis of SLC39A8 and its correlated metabolic DEGs. The screened hub genes, NSDHL, ACLY, ACAT2, and FASN may be downstream target genes of SLC39A8. However, further study about PH is required for a better understanding of the role of ZIP8 in PH.

## ACKNOWLEDGEMENTS

We really appreciate GEO databases for providing platforms and contributors for uploading meaningful datasets, and Xiantao online tools for mapping.

### Funding

This project was supported by funding from the National Natural Science Foundation of China (grant. nos. 82000062 and 81960015), the Science Foundation for distinguished Young Scholars of Jiangxi Province (grant no. 20212ACB216008), the Young Talents Project Foundation of Science and Technology Department of Jiangxi Province (grant no. 20204BCJ23020), the Natural Science Foundation of Henan Province (22A180010), Research Foundation of Xinxiang Medical University (XYBSKYZZ202108) and the Key Research and Development Program of Jiangxi Province (20192BBG70012), the Research Foundation of College student innovation project of Henan (202210472019). The funders had no role in study design, data collection and analysis, decision to publish, or preparation of the manuscript.

### Grant Disclosures

The following grant information was disclosed by the authors:
National Natural Science Foundation of China: 82000062, 81960015.
Science Foundation for distinguished Young Scholars of Jiangxi Province: 20212ACB216008.
Young Talents Project Foundation of Science and Technology Department of Jiangxi Province: 20204BCJ23020.
Natural Science Foundation of Henan Province: 22A180010.
Research Foundation of Xinxiang Medical University: XYBSKYZZ202108.
Key Research and Development Program of Jiangxi Province: 20192BBG70012.
Research Foundation of College student innovation project of Henan: 202210472019.

### Competing Interests

The authors declare there are no competing interests.

### Author Contributions

- FanRong Zhao analyzed the data, prepared figures and/or tables, and approved the final draft.
- Yujing Chen analyzed the data, prepared figures and/or tables, and approved the final draft.
- Yuliang Xie performed the experiments, prepared figures and/or tables, and approved the final draft.

- Shuang Kong performed the experiments, prepared figures and/or tables, and approved the final draft.
- LiaoFan Song performed the experiments, prepared figures and/or tables, and approved the final draft.
- Hanfei Li performed the experiments, prepared figures and/or tables, and approved the final draft.
- Chao Guo analyzed the data, prepared figures and/or tables, and approved the final draft.
- Yanyan Yin analyzed the data, prepared figures and/or tables, and approved the final draft.
- Weifang Zhang conceived and designed the experiments, authored or reviewed drafts of the article, and approved the final draft.
- Tiantian Zhu conceived and designed the experiments, authored or reviewed drafts of the article, and approved the final draft.

## Animal Ethics

The following information was supplied relating to ethical approvals (i.e., approving body and any reference numbers):

Laboratory Animal Center of Henan Province, Xinxiang Medical University

## Data Availability

The raw data is available in the Supplemental Files.

The microarray data are available at GEO: GSE24988, GSE113439, GSE117261, GSE15197.

## Supplemental Information

Supplemental information for this article can be found online at http://dx.doi.org/10.7717/peerj.15939#supplemental-information.

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
