# Peer review of "Identification of Zip8-correlated hub genes in pulmonary hypertension by informatic analysis"

_PeerJ, doi:10.7717/peerj.15939_

## Round 0.1 · original submission · Major Revisions

I suggest authors to go through all the comments from the reviewer and address them in the revised version.

·

Basic reporting

The article is clear and written in professional English.
Sufficient review of literature was presented to provide a clear the background.
Problem and objectives were described clearly.
Article manifests professional structure of a typical research article furnished with the relevant results supported with the relevant data (tables, figures and supplementary data regarding databases and the corresponding bioinformatic software used).

Experimental design

This research article presents original and primary research which lie within the aims and scope of the Peerj Journal.
Research question is well defined and gap in the existing knowledge filled with by the current research clearly reported.
Ethical and technical standards were adopted to perform the research. I have reviewed the permissions/license required to perform these experiments on animals (mice) by the authors.
Methods described by the authors are quite sufficient and within the scope/limit of the planned research.

Validity of the findings

The results reported by the authors are novel and meaningful.
All necessary data pertaining to results have been provided.
The results presented are created using sufficient statistical and bioinformatic analyses
Conclusions are well stated and are linked to the original research performed.
The conclusions are in line with the original research question and results.

Additional comments

Though an extensive metabolomic analysis specially the transcriptomics and proteomics in vivo were lacking however, further research would be helpful beyond the current reporting.
Authors are encouraged to do further research using big data sets in future as suggested.

·

Basic reporting

Some references missing (Please check additional comments)

The figure quality should be improved

Experimental design

The criterion used in some of the methods is inappropriate, hindering the validation of the results and raising concerns about the validity and reliability of the findings.

Validity of the findings

The utilization of inappropriate criteria in various methods raises concerns about the validity and reliability of the findings. It is crucial for the authors to address these concerns and provide supporting references to justify their conclusion.

Additional comments

In this work, the authors aim to apply different bioinformatics tools including DE analysis, GSEA, and PPI network analysis on both human and mice data to unravel the role of SLC39A8 in the progression of pulmonary hypertension. However, the reviewer feel that the results obtained from this study are based on poor assumption. And some issues need to be addressed to improve the manuscript.

1) Line 35, please provide the full name for "NC" on line 35, as it is the first occurrence of this abbreviation in the text.
2) Line 58, “Currently, drugs mediates …”, change “mediates” to “mediate”.
3) Line 59, change semicolon to comma.
4) Line 68-71, I suggest authors to remove the description about ZIP12 here since the whole paragraph is talking about the role of ZIP8.
5) Line 73-74, “In this study…”, I suggest authors to start a new paragraph to briefly introduce the workflow implemented in this study. This will help readers understand the overall methodology and approach used in the research.
6) Line 80, “R package (version 4.2.1)”, it should be R instead of R package, and in line 134, “R 3.5.1”, is different versions of R used here?
7) Missing version information regarding R package, for example, “sva”, “limma”, “ggplot2”, please provide the version information here.
8) Line 122, please provide the full name for “NES” here since it’s its first appearance.
9) Line 122, P.adj < 0.05, FDR (q value) < 0.25, I am quite confusing here. q value is one type of adjusted p-value, what is the p.adj here? And the cutoff (0.25) used here for FDR adjusted p-value is a little bit high here. Also, usually |log2FC| is used as one of the criterions for selecting the DE genes, authors should address the rationale behind not including |log2FC| in their analysis and provide an explanation for their alternative approach.
10) Line 128, “… combined score of > 0.4 was considered statistically significant”, can authors explain why 0.4 is used here? Any reference to support this?
11) Line 131, “Hub genes were … MMC, Degree and Closeness were selected.” Please provide references for these methods.
12) Line 133, for section “Statistical Analysis”, what’s the input for these different tests? DEGs from DE analysis? Or all the genes? Please make it clear.
13) Line 136, “… between two PH and NC”, remove “two” here.
14) Line 141, section “SLC39A8 was lowly expressed in PH”, the section title is inappropriate here since most of section discuss the DE analysis and GSEA based on the merged dataset, only the last several sentences discuss the SLC39A8.
15) Line 152, “3 of the top 5”, should be “4 of the top 5”.
16) Line 155, “We next analyzed the expression of SLC39A8 …”, including a discussion about whether SLC39A8 is among the differentially expressed genes (DEGs) and if it is the top DEG can provide valuable insights.
17) Line 158, “Slc39a8”, please change it to “SLC39A8” to keep consistent.
18) Line 164, “Then 6083 SLC39A8-correlatted were…”. Typo, should be correlated, and should add “genes” after “correlated”. Moreover, there seems to be confusion regarding the number of correlated genes obtained. If the previous section states that only 5228 DEGs were obtained, it is necessary to address how the number increased to 6083 after the filtering process.
19) Line 174-176, “Among them, CLU …which indirectly suggests SLC39A8 may play a …”. Usually, correlation > 0.7 is defined as high correlation. For here, the two selected genes both show correlation < 0.7, so it’s hard to support the conclusion here based on this correlation analysis. And can authors find any reference to support this conclusion?
20) Line 180-183, repetitive information.
21) Line 200-202, “Based on these results…”, is there any reference to support this conclusion?
22) Line 204, “PAH”, please provide the full name here.
23) Line 267-269, can authors find any reference to support the finding in this study?
24) Figure quality should be improved.
25) Figure 1,

Reviewer 3 ·

Basic reporting

In Abstract Method, “all differentially expressed genes (DEGs) between PH and NC from the Biotechnology Gene Expression Omnibus (NCBI GEO) database.”, please clarify what NC stands for in the Abstract since this is the first time that this terminology appears in this paper.

To enhance transparency, it would be beneficial for the authors to present a supplementary table that includes Limma output on the DEGs identified in the study (estimates, p-values/adjusted p-values/FDR, log2FoldChange values, etc.). By providing this additional information, readers will have access to the detailed results and be able to assess the statistical significance and magnitude of gene expression changes more effectively.

Experimental design

For Mouse model of PH/Animal Experiment, please disclose the statistical power for the statistical analysis. Specifically, a statistical power should be calculated for the comparison in expression levels of Slc39a8 in the lungs of normoxia and Su/Hx treated PH mice (n=10).

Validity of the findings

The results obtained in this study are promising.

Additional comments

This study identified four genes whose expression was down-regulated in PH mice, such as Fasn, Nsdhl, Acat2 and Acly. Among these four genes, Fasn and Acly are key fatty acid synthases, Nsdhl is involved in cholesterol synthesis, and Acat2 is involved in the metabolic conversion of cholesterol. These results provide new insights into the role of Zip8 in PH.

Reviewer 4 ·

Basic reporting

Please see Additional comments.

Experimental design

Please see Additional comments.

Validity of the findings

Please see Additional comments.

Additional comments

The authors combined bioinformatics and qPCR to explore the role of ZIP8 (aka SLC39A8) in the pathogenesis and prognosis of pulmonary hypertension. Overall, this study is suitable for publication, only if the authors address the following issues:

1. Throughout the manuscript, it seems better to use Grammarly (https://www.grammarly.com/) to check & correct potential grammatical errors or typos. For example,
1.1 It would be more readable to add a blank line after every paragraph.
1.2 In Data Collection and Processing of MATERIALS & METHODS, it seems better to change "Of these datasets, only those containing the expression of SLC39A8 PH patients and normal control (NC) samples were selected, such as GSE24988 (62 PH and 22 NC), GSE113439 (15 PH and 11 NC), GSE117261 (58 PH and 25 NC) and GSE15197 (18 PH and 13 NC)" into "Of these datasets, only those containing the expression of SLC39A8 in both PH patient and normal control (NC) samples were selected, include ...".
1.3 In Identifying SLC39A8-correlated DEGs of RESULTS, it seems better to change "Then 6083 SLC39A8-correlatted were identified when considered p.adj<0.05" into "Then 6083 SLC39A8-correlated genes were identified when considered p.adj<0.05", which would be more accurate.
1.4 In GSEA analysis of SLC39A8-correlated DEGs, it seems better to change "we concluded that the role of SLC39A8 in the progression of PH may attributed to cholesterol and/or steroid metabolism" into "we concluded that the role of SLC39A8 in the progression of PH may be attributed to cholesterol and/or steroid metabolism", which would be more accurate.

2. In all FIGURES, it would be clear and more readable to expand on figure legends by explaining the meanings of colors, groups, lines, and abbreviations. For example,
2.1 In the legend of Figure 1, it seems better to change "(C) GSEA analysis shows enriched KEGG pathways. (D) GSEA analysis shows enriched Reactome pathways" into "(C) GSEA analysis shows KEGG pathways enriched for the DEGs. (D) GSEA analysis shows Reactome pathways enriched for the DEGs", which would be clearer. In addition, please mention the implications of a positive or negative NES score.
2.2 In the legends of all Figures, it would be more rigorous to mention the sample size.

These revisions would greatly help readers, who do not specialize in bioinformatics, to understand the results and their implications easily and efficiently.

3. In ABSTRACT:
3.1 In Background, it seems better to change "However, the role of Zip8 in the progression of PH remains unclear" into "However, the role of Zip8 in PH remains unclear", which would be clearer and more accurate. Because "progression" seems to specifically refer to the advancement or worsening of a disorder — the advanced stage, rather than the onset.
3.2 In Methods, it seems better to change "Bioinformatics analysis was employed to identify the expression of SLC39A8 expression, all differentially expressed genes (DEGs) between PH and NC from the Biotechnology Gene Expression Omnibus (NCBI GEO) database" into "Bioinformatics analysis was employed to identify SLC39A8 expression patterns and differentially expressed genes (DEGs) between PH patients and normal controls (NC), based on four datasets (GSE24988, GSE113439, GSE117261, and GSE15197) from the Biotechnology Gene Expression Omnibus (NCBI GEO) database", which would be clearer and more informative.
3.3 In Methods, it seems better to change "Gene set enrichment analysis (GSEA) was performed to analyze the enriched signaling pathways" into "Gene set enrichment analysis (GSEA) was performed to analyze signaling pathways enriched for DEGs", which would be clearer and more cohesive (that is, closely linked to the sentences before & after it).
3.4 In Methods, it would be clearer to mention how the authors identify (or define) BOTH "SLC39A8-correlated genes" — an expression frequently mentioned in this manuscript AND "SLC39A8-correlated differentially expressed genes (DEGs)" — an expression mentioned in Results.
3.5 In Results, it seems better to change "The expression of SLC39A8 expression was downregulated in PH patients, and this variation was validated in PH (SU5416/Hypoxia) mice lung tissue" into "SLC39A8 expression was downregulated in PH patients, and this expression pattern was validated in PH (SU5416/Hypoxia) mouse lung tissue", which would be clearer and easier to understand.
3.6 In Results, it seems better to change "Seven hub genes of SLC39A8-correlated metabolic DEGs were identified, and the expression of these genes were analyzed in PH patients and controls, and further validated in PH mice" into "Within these SLC39A8-correlated metabolic DEGs, seven genes were identified hub genes. The expression patterns of hub genes were analyzed between PH patients and controls and further validated in PH mice", which would be clearer and more cohesive.
3.7 In Results, it seems better to change "Finally, four genes expression were downregulated in PH mice, such as Fasn, Nsdhl, Acat2, and Acly" into "Finally, four genes (Fasn, Nsdhl, Acat2, and Acly) were downregulated in PH mice", which would be more concise.
3.8 In Results, it seems better to change "Of the 4 genes, Fasn and Acly are the key fatty acids synthesis enzymes, Nsdhl is involved in cholesterol synthesis, while Acat2 is involved in cholesterol metabolic transformation" into "Of the 4 genes, Fasn and Acly are key enzymes in fatty acids synthesis, Nsdhl is involved in cholesterol synthesis, and Acat2 is implicated in cholesterol metabolic transformation", which would be clearer and coordinated (that is, a parallel structure).
3.9 In Results, it seems to change "These results provide novel insight into the role of Zip8 in PH" into "Taken together, these results provide novel insight into the role of Zip8 in PH", which would be clearer and easier to read.
3.10 In Results, the authors explained four out of the seven hub genes, but it could be more rigorous to mention the expression patterns of the remaining three genes. Alternatively, please explain why the authors did not explore the three genes.

4. In INTRODUCTION:
4.1 In Paragraph 1, it seems better to change "Pulmonary hypertension (PH) is a syndrome characterized by marked remodeling of the pulmonary vasculature and increased pulmonary vascular resistance and pressure , which ultimately leads to right heart failure and even death" into "Pulmonary hypertension (PH) is a syndrome characterized by marked remodeling of the pulmonary vasculature and increased pulmonary vascular resistance and pressure, leading to right heart failure and even death", which would be clearer and more concise.
4.2 In Paragraph 1, it seems better to change "PH pathogenesis is multifactorial and is presented as an aberrantly elevated pulmonary artery pressure (PAP) and a persistent increase in pulmonary vascular resistance and vascular remodeling" into "PH pathogenesis is multifactorial and presented as the aberrantly elevated pulmonary artery pressure (PAP), the persistent increase in pulmonary vascular resistance, and the vascular remodeling", which would be clearer and more coordinated.
4.3 In Paragraph 1, it seems better to change "There are several pathological features of the disorder, such as remodeling of the distal pulmonary vasculature, and infiltration of inflammatory cells, extension of the pulmonary artery smooth muscle cells (PASMC) into typically nonmuscularized vessels" into "Its pathological features include remodeling of the distal pulmonary vasculature, infiltration of inflammatory cells, and extension of the pulmonary artery smooth muscle cells (PASMC) into typically nonmuscularized vessels", which would be clearer and easier to read.
4.4 It would be more coherent and cohesive to integrate Paragraph 2 ("Currently, drugs ... exercise tolerance") into Paragraph 1. Similarly, it seems better to change "Currently, drugs mediates improvement of vascular relaxation and inhibition of cell proliferation has shown favorable results; but the disease is progressive and the long term results are far from ideal" into "PH has been mitigated by drugs that improve vascular relaxation and inhibit cell proliferation has shown favorable results, but the long term results of these drugs are far from ideal", which would be more cohesive and easier to read. In addition, it would be more rigorous to cite references to support the statements in Paragraph 2.
4.5 In Paragraph 3, it seems better to change "SLC39A8 encodes a zinc transporter ZIP8, a member of ZIPs, whose expression was found to be highest in kidney, lung, and testis[3], and relatively more abundantly in endothelium" into "SLC39A8 encodes a zinc transporter ZIP8, a member of ZIPs, whose expression was found to be highest in kidney, lung, and testis, compared with other organs[3]; in different organs, its expression was abundant in endothelium[3]", which would be clearer and easier to understand.
4.6 In Paragraph 3, it seems better to change "The important role of ZIP8 in the lung has been demonstrated in several studies, for example" into "The important role of ZIP8 in the lung has been demonstrated in several studies. For example", which would be easier to understand.
4.7 In Paragraph 3, to take advantage of the evidence "There were also other researches that suggested that the expression of ZIP12 ( another member of ZIPs) was induced in the vasculature in human patients and rat models of PH in vivo [8-11], which was at least partially responsible for hypoxia-induced PH in both human and rats", it would be more rigorous and logical to cite references supporting that "member of ZIPs" could play a similar role in (lung) diseases. Such references would justify the authors' hypothesis that ZIP12's role in PH could suggest the importance of ZIP8 in PH.
4.8 In Paragraph 3, it seems better to change "Based on the above research background, we reasoned that ZIP8, hereafter referred as SLC39A8, also plays a vital role in the progression of PH" into "Based on the above research background, we hypothesize that ZIP8, hereafter referred as SLC39A8, could play a vital role in the pathogenesis of PH", which would be easier to understand and more accurate.
4.9 In Paragraph 3, it seems better to change "In this study, we analyzed SLC39A8 expression in PH patients and mice, and explored the role of SLC39A8 in the progression of PH and the potential mechanism involved" into "To test this hypothesis, we analyzed SLC39A8 expression in PH patients as well as mice and predicted SLC39A8-related molecular mechanisms in PH pathogenesis", which would be clearer and more cohesive.

5. In MATERIALS & METHODS:
5.1 In Data Collection and Processing, it seems better to change "Using the keywords “pulmonary hypertension” and “Homo sapiens”, 7 datasets were screened out. The original data were obtained from the National Center for Biotechnology Gene Expression Omnibus (NCBI GEO) database" into "We searched the National Center for Biotechnology Gene Expression Omnibus (NCBI GEO) database for datasets related to both “pulmonary hypertension” and “Homo sapiens”, identifying 7 datasets", which would be clearer and easier to understand.
5.2 In Identifying Differentially Expressed Genes, it would be more rigorous to mention the fold change value that was used to identify the DEGs.
5.3 In Gene Set Enrichment Analysis (GSEA) of DEGs, it seems better to change "Gene set enrichment analysis (GSEA) was performed using GSEA/MsigDB (http:// www.broadingstitute.org/gsea/msigdb/index.jsp) to analyze the DEGs" into "Gene set enrichment analysis (GSEA) was performed on the DEGs using GSEA/MsigDB (http:// www.broadingstitute.org/gsea/msigdb/index.jsp)", which would be more accurate and rigorous. In addition, it would be more informative and easier to understand to explain the implications of NES (that is, what is meant by a positive or negative NES score).

6. In RESULTS:
6.1 It would be clearer to end each paragraph in RESULTS with one sentence: "Together, these results suggest that ..." (a pattern like PMID: 34715879, PMID: 34384362, PMID: 35965679, and PMID: 34537192), summarizing a paragraph AND highlighting the implications of all results in the paragraph.
6.2 In SLC39A8 was lowly expressed in PH, it seems better to change "As shown in Figure 1A, volcano plot showed a total of 5228 DEGs (padj<0.05) were identified from the merged datasets" into "As shown in Figure 1A, a total of 5228 DEGs (padj<0.05) were identified from the merged datasets (GSE24988, GSE113439, GSE117261, and GSE15197)", which would be more concise and informative.
6.3 In SLC39A8 was lowly expressed in PH, it seems better to change "Enriched Kyoto Encyclopedia of Genes and Genomes (KEGG) pathways from the GSEA analysis of all DEGs were shown in Figure 1C, the top 5 pathways such as “Arrhythmogenic Right Ventricular Cardiomyophthy Arvc”[13], “Ecm Receptor Interaction”[14, 15], “Wnt Signaling Pathway” [16, 17], “Hypertrophic Cardiomypathy Hcm”[18, 19], “Focal Adhesion”[20-22], “Dilated Cardiomyopathy”[23, 24] are all associated with PH" into "Based on the GSEA analysis of all DEGs, top 6 Kyoto Encyclopedia of Genes and Genomes (KEGG) pathways (Figure 1C) included “Arrhythmogenic Right Ventricular Cardiomyophthy Arvc”[13], “Ecm Receptor Interaction”[14, 15], “Wnt Signaling Pathway” [16, 17], “Hypertrophic Cardiomypathy Hcm”[18, 19], “Focal Adhesion”[20-22], and “Dilated Cardiomyopathy”[23, 24], all of which could be associated with PH", which would be easier to understand and more cohesive.
6.4 In SLC39A8 was lowly expressed in PH, it seems better to change "On the other hand, 3 of the top 5 enriched Reactome pathways from the GSEA analysis (Figure 1D) such as “Degradation of the Extracellular Matrix”[25, 26], “Cilium Assembly”, “Extracellular Matrix Organization”[25] and “Signaling By Tgfb Family Members”[27] are also involved in the pathogenesis of PH" into "On the other hand, 4 of the top 5 enriched Reactome pathways (Figure 1D) encompassed “Degradation of the Extracellular Matrix”[25, 26], “Cilium Assembly”, “Extracellular Matrix Organization”[25], and “Signaling By Tgfb Family Members”[27], all of which could be involved in the pathogenesis of PH", which would be more coherent and rigorous.
6.5 In Identifying SLC39A8-correlated DEGs, it seems better to change "we performed correlation analysis among all DEGs and SLC39A8 using Pearson correlation coefficients" into "we used Pearson correlation coefficients to perform correlation analysis on all DEGs, thereby identifying the DEGs with expression patterns similar to SLC39A8 expression", which would be more accurate and easier to understand.
6.6 In GSEA analysis of SLC39A8-correlated DEGs, it would be more concise to delete "using gene set collections from the MsigDB (https://www.gseamsigdb.org/gsea/msigdb/collections.jsp). ClusterProfiler[4.4.4] was used for GSEA analysis, and the c2.cp.all.v2022.1.Hs.symbols.gmt [All Canonical Pathways](3050) was selected as the reference gene set. P.adj<0.05 and false discovery rate (FDR, qvalue) <0.25 were used to indicate significant enrichment score".
6.7 In Identifying key SLC39A8-correlated metabolic DEGS between PH and NC, it would be more rigorous and easier to understand to explain why the Venn diagram had 1600 (transcripts of metabolic genes), rather than 1,660 (human genes assigned to 86 metabolic pathways).
6.8 In Identifying key SLC39A8-correlated metabolic DEGS between PH and NC, it seems better to change "The SLC39A8-correlated metabolic DEGs showed a significant cluster of interactions and networks. And the PPIs with combined scores greater than 0.4 were obtained for constructing networks by using the STRING database" into "The 202 SLC39A8-correlated metabolic DEGs were analyzed by STRING database to explore PPI, in which interactions with a combined score greater than 0.4 were obtained to construct PPI networks", which would be more informative and easier to understand.
6.9 In Identifying key SLC39A8-correlated metabolic DEGS between PH and NC, it would be more informative to list the names of the "7 common genes".
6.10 In Validation of key SLC39A8-correlated metabolic DEGs in a mice model Using RTPCR Analysis, it seems better to change "the expression of 6 genes were low in PH patients, such as cholesterol acyl-transferase 2 (ACAT2), NAD(P)-dependent steroid dehydrogenase-like (NSDHL), farnesyl diphosphate synthase (FDPS), fatty acid synthase (FASN), ATP-citrate lyase (ACLY) and Acetyl-CoA synthetase 2 (ACSS2)" into "PH patients lowly expressed 6 genes: cholesterol acyl-transferase 2 (ACAT2), NAD(P)-dependent steroid dehydrogenase-like (NSDHL), farnesyl diphosphate synthase (FDPS), fatty acid synthase (FASN), ATP-citrate lyase (ACLY) and Acetyl-CoA synthetase 2 (ACSS2)", which would be clearer and easier to read.
6.11 In Validation of key SLC39A8-correlated metabolic DEGs in a mice model Using RTPCR Analysis, it would be clearer to rewrite and refine the sentence "While the expression of FDFT1 did not differ between PH patients and NC group. It is possible that the actual differences were small and no significance when differential analysis is statistically different with a high p value, which attributed to different statistical methods."

---

## Round 0.2 · Minor Revisions

I suggest authors to go through the comments from the reviewers and address them in the final version of the manuscript

·

Basic reporting

The article is clear and written in professional English.
Sufficient review of literature was presented to provide a clear the background.
Problem and objectives were described clearly.
Article manifests professional structure of a typical research article furnished with the relevant results supported with the relevant data (tables, figures and supplementary data regarding databases and the corresponding bioinformatics software used).

Experimental design

This research article presents original and primary research which lie within the aims and scope of the Peerj Journal.
Research question is well defined and gap in the existing knowledge filled with by the current research clearly reported.
Ethical and technical standards were adopted to perform the research. I have reviewed the permissions/license required to perform these experiments on animals (mice) by the authors.
Methods described by the authors are quite sufficient and within the scope/limit of the planned research.

Validity of the findings

The results reported by the authors are novel and meaningful.
All necessary data pertaining to results have been provided.
The results presented are created using sufficient statistical and bioinformatic analyses
Conclusions are well stated and are linked to the original research performed.
The conclusions are in line with the original research question and results.

Additional comments

As suggested, metabolomic analysis are planned by authors to determine the mechanism of Zip8 in the progression of PH which is appreciated.

·

Basic reporting

Check additional comments

Experimental design

Check additional comments

Validity of the findings

Check additional comments

Additional comments

The authors have addressed most of my comments, but there are still several minor comments that need to be addressed before the manuscript can be accepted:

1) Line 32, “… was abundantly in xxx”, grammar error, change abundantly to abundant. Also, please check the manuscript for any other grammar errors.
2) Line 39, “Omnibus (NCBI GEO) database)”, remove “)” after database.
3) Line 62, “It pathological features of the disorder xxx”, typo? Change “It” to “The”.
4) Line 109, “sva(v332.10)”, “limma(v3.24.4)”; Line 116, “ggplot2[3.3.6]”. Please use a consistent format for the version.
5) Line 161, “PPI network was constructed by Cytoscape,”, change “,” to “.”.

Reviewer 3 ·

Basic reporting

The revised manuscript was clearly written. Literature references were fully provided. The structure of the manuscript looks good.

Experimental design

The study design makes sense in the revised manuscript. Research question was well defined. The appropriate methods and statistical analysis were used and conducted. Enough details related to the study design were provided.

Validity of the findings

The results and conclusion in the revised manuscript are relatively promising. The statistical analysis results were provided to prove the validity of the findings. The conclusions are well stated and closely linked to the research question.

Additional comments

Sine the manuscript was revised based on reviewers' comments. I recommend to accept this paper.

Reviewer 4 ·

Basic reporting

Thank the authors for their efforts to respond to all of my comments. Overall, this version would be suitable for publication, only if the authors address the following issues, to which the authors did not seem to adequately respond:

1. My previous comment 2.2 (2.2 In the legends of all Figures, it would be more rigorous to mention the sample size.). The authors have not mentioned the sample size in Figure 2 and 5.

Experimental design

N/A

Validity of the findings

N/A

Additional comments

N/A

---

## Round 0.3 · accepted · Accept

Your manuscript at this stage is acceptable for publication.